# Genome Analysis of *Thinopyrum intermedium* and Its Potential Progenitor Species Using Oligo-FISH

**DOI:** 10.3390/plants12213705

**Published:** 2023-10-27

**Authors:** Fei Qi, Shuang Liang, Piyi Xing, Yinguang Bao, Richard R.-C. Wang, Xingfeng Li

**Affiliations:** 1State Key Laboratory of Crop Biology, Shandong Agricultural University, Tai’an 271018, China; 2021010020@sdau.edu.cn (F.Q.); pyxing@sdau.edu.cn (P.X.); ygbao@sdau.edu.cn (Y.B.); 2Agronomy College, Shandong Agricultural University, Tai’an 271018, China; 3USDA-ARS Forage & Range Research Laboratory (FRRL), Logan, UT 84322-6300, USA

**Keywords:** Triticeae, genome constitution, FISH, evolution, karyotype

## Abstract

The genome composition of intermediate wheatgrass (IWG) is complex and continues to be a subject of investigation. In this study, molecular cytogenetics were used to investigate the karyotype composition of *Th. intermedium* and its relative diploid species. St_2_-80 developed from *Pseudowroegneria strigose* and pDb12H developed from *Dasypyrum breviaristatum* were used as probes in fluorescence in situ hybridization (FISH) to classify the chromosomes of *Th. intermedium* into three groups, expressed as **J^vs^J^vs^J^r^J^r^StSt**. A combined multiplex oligonucleotide probe, including pSc119.2-1, (GAA)_10_, AFA-3, AFA-4, pAs1-1, Pas1-3, pAs1-4, and pAs1-6, was used to establish the FISH karyotype of ten accessions of *Th. intermedium*. Variability among and within the studied accessions of intermediate wheatgrass was observed in their FISH patterns. Results of this study led to the conclusions that **J^vs^** had largely been contributed from *Da. breviaristatum*, but not the present-day *Da. villosum*; IWG had only one **J** genome, **J^r^**, which was related to either *Th. elongatum* or *Th. bessarabicum*; and **St** was contributed from the genus *Pseudoroegneria* by hybridization with *Th. junceiforme* or *Th. sartorii*.

## 1. Introduction

Intermediate wheatgrass (IWG; *Thinopyrum intermedium* (Host) Barkworth & D.R. Dewey; 2n = 6x = 42) is an important worldwide forage crop and a valuable gene reservoir for wheat improvement [1,2,3]. It has contributed many desirable traits such as resistance to three wheat rust diseases (leaf, stem, and stripe rust), barley yellow dwarf virus (BYDV), and wheat streak mosaic virus (WSMV), and grain quality for wheat improvement [3,4,5,6,7,8]. The IWG has also been developed into a perennial grain crop, named “Kernza” [9,10,11]. Therefore, IWG had been the subject of investigations by numerous researchers for a long time.

The genome constitution of IWG had been proposed as **E_1_E_2_X [12]**, **J^e^J^e^S** [12] (while the symbol **S** was later changed to **St** [13]). The presence of the **St** genome of *Pseudoroegneria* in IWG was substantiated by all subsequent studies [14,15,16,17]. Using **St** genomic DNA as a probe, there were eight to ten chromosomes showing the signals at centromeres [14,15,16,18,19], which was given the symbol **J^s^** [15] even though it was not a complete set of 14 chromosomes. Later, St_2_-80 was developed as a new FISH marker for the **St** genome, useful in genome analysis of Triticeae species and hybrids [20]. When genomic DNA of *Dasypyrum villosum* or an oligo DNA from *D. breviaristatum* (H. Lindb.) Fred. (syn. *Dasypyrum hordeaceum* (Hack.) P. Candargy) pDb12H [21,22] was used as a probe, fourteen chromosomes, including those eight to ten **J^s^** type chromosomes, showed hybridization signals. On the other hand, the 14 **J** chromosomes were distinguished from **J^s^** and **St** chromosomes by a long terminal repeat (LTR) sequence pMD232-500, from *Secale* [18]. Therefore, **J^s^** was changed to **J^vs^** and **J** was changed to **J^r^**, and **J^r^J^r^J^vs^J^vs^StSt** was proposed to be the genome symbol for *Th. intermedium* [23]. Hence, this genome constitution had been adopted by other researchers [24,25]. It should be noted that **J^r^** was proposed as an ancestral genome to the present-day **J^b^** and **J^e^** genome in the diploid species *Th. bessarabicum* (Savul. and Rayss) A. Löve and *Th. elongatum* (Host) D. Dewey, respectively [23], based on the relationships between **R** and the **J** genomes revealed by restriction site differences [26].

Genetic resources for IWG have been generated, including EST-SSR markers [15], the first consensus genetic map using genotyping-by-sequencing [27], QTL mapping [28], draft genome sequence [29], and STS marker sets for the three genomes of IWG [30]. Molecular markers for disease resistance were also developed and located on various IWG chromosomes [31,32,33].

Genomic in situ hybridization (GISH) had been a valuable cytogenetic technique widely used to determine genome constitutions of plant species. However, the oligonucleotide fluorescence in situ hybridization (oFISH) could pinpoint the chromosomal locations of known genes whose DNA sequences were used as probes.

Karyotypes of oligonucleotide fluorescence in situ hybridization, coupled with molecular markers, would be useful for aiding the precise identification of individual chromosomes in IWG. Using dual and multiplex oligonucleotide FISH, we studied ten accessions of IWG having widely different origins along with IWG-related or progenitor species of the ploidy ranging from diploid to hexaploid. The objectives are (1) to test if pDb12H and St_2_-80 are sufficient to distinguish **J^r^**, **J^vs^**, and **St** in *Thinopyrum* species, (2) to determine if variability in IWG can be visualized with oFISH, and (3) to refine the relationships between IWG and its progenitor diploid species. Results will provide a clear picture of the relationships among studied species in the tribe Triticeae.

## 2. Results

### 2.1. OligoFISH of Ten Accessions of IWG

In our study, chromosomes in ten accessions of IWG (Table 1) were probed with two oligonucleotides, pDb12H and St_2_-80, to distinguish the **J^r^**, **J^vs^**, and **St** genomes. Then, the same chromosome spreads were probed with bulked oligonucleotides consisting of pSc119.2-1, (GAA)_10_, AFA-3, AFA-4, pAs1-1, Pas1-3, pAs1-4, and pAs1-6 after the slides were cleaned of the pDb12H and St_2_-80 probes.

The probe pDb12H labeled 14 chromosomes of the **J^vs^** genome with green color on both arms, whereas the probe St_2_-80 gave red-colored signals at distal ends of 14 chromosomes, each of the **J^vs^** and **J^r^** genomes, and in the interstitial regions of 14 short chromosomes of the **St** genome (Figure 1).

All ten accessions of IWG had the same FISH pattern from the two probes pDb12H and St_2_-80, showing their common genome constitution **J^r^J^r^J^vs^J^vs^StSt** (left side of Figure 2). However, the multiplex oligonucleotide FISH patterns revealed variable signals on some chromosomes of the **J^vs^** and **J^r^** genomes among accessions, or even within an accession (right side of Figure 1 and Figure 2).

### 2.2. OligoFISH of Three Tetraploid Species of Thinopyrum Genus

Both *Th. junceiforme* and *Th. sartorii* had the **J^vs^J^vs^J^r^J^r^** genome constitution, as shown in Figure 3 and Figure 4, respectively. Variation in FISH signals between homologous chromosomes could be detected. On the other hand, *Th. scirpeum* was found to be an autotetraploid having the **J^r^J^r^J^r^J^r^** genome composition (Figure 5). The former two species differ from *Th. intermedium* by lacking the **St** genome.

### 2.3. OligoFISH of Four Diploid Species That Were Implicated as Progenitors of IWG

Three diploid species, *Th. elongatum, Th. bessarabicum,* and *Ps. sspicata*, were studied using the two probes pDb12H and St_2_-80 and multiplex oligos pSc119.2-1, (GAA)_10_ (green), AFA-3, AFA-4, pAs1-1, Pas1-3, pAs1-4, and pAs1-6 (red) (Figure 6). None of them showed strong green signals from pDb12H, indicating these species could not be the donor of the **J^vs^** genome in IWG. Some chromosomes of *Th. elongatum* had very faint green signals in the interstitial regions, whereas no such signals were detected in *Th. bessarabicum* (Figure 6A–D). Weak green signals of the pDb12H probe were detected at the centromere region of eight chromosomes in *Da. Villosum* probed by pDb12H and St_2_-80 (Appendix A).

Interestingly, the red signal of St_2_-80 occurring in *Th. elongatum, Th. bessarabicum,* and *Ps. spicata* differed in its distribution on chromosomes. This red color was intense and located on distal ends of chromosomes in the two diploid *Thinopyrum* species. It was mostly spread out in the interstitial regions of *Pseudorogneria* chromosomes. There were more chromosomes with the red hybridization signal in *Th. bessarabicum* (Figure 7) than *Th. elongatum* (Figure 6).

### 2.4. In Situ Hybridization of Th. intermedium Using Genomic DNA of Th. elongatum and Th. bessarabicum, and Oligo Probes pDb12H and St_2_-80

When *Th. intermedium* was probed with genomic DNA of *Th. elongatum* and *Th. bessarabicum*, and the two oligos pDb12H and St_2_-80, it was found that the genomic DNA of both diploid *Thinopyrum* species hybridized the same 14 chromosomes (Figure 7A,B). This result indicated that IWG had only one **J** genome. On the other hand, the oligo probes pDb12H and St_2_-80 hybridized the other 28 chromosomes, 14 each of the **J^vs^** and **St** genome, respectively (Figure 7C).

### 2.5. In Situ Hybridization of Th. intermedium Using Genomic DNA of Dasypyrum villosum and Oligo Probes pDb12H and St_2_-80

Genomic DNA of *Da. villosum* and oligo probes pDb12H and St_2_-80 were used to hybridize the same chromosomes of IWG in a root-tip cell (Figure 8). Both **V** genomic DNA (Figure 8A) and pDb12H (Figure 8B) hybridized the same 14 chromosomes in *Th. intermedium*. The St_2_-80 probe hybridized all 42 chromosomes, but at different sites and at a different intensity. Chromosomes showing green signals of pDb12H had the red signals from St_2_-80 at telomeric ends. Intense red signals were on 14 short chromosomes, and another 14 chromosomes had red signals at the ends of chromosome arms that embraced bluish interstitial segments.

## 3. Discussion

### 3.1. Prior Studies on Thinopyrum intermedium and Related Species

The genome constitution of IWG had been investigated by many research groups and its genome symbol changed over the years, as described by Wang and Lu (2014) [34]. The genome constitution of IWG had been designated as **E_1_E_2_X [12]**. Using the methods of chromosome karyotyping, Giemsa C-banding, and meiotic pairing in hybrids, Liu and Wang [35] concluded that the X genome in *Th. intermedium* is the **S** genome from an unspecified *Pseudoroegneri* pecies. Three tetraploid species, *Elytrigia caespitosa, Lophopyrum nodosum, Pseudoroegneria geniculata* ssp. *scythica,* were determined to have the **J^e^S** genome constitution in the same study. Then, the symbol **S** was changed to **St** in 1995 [13]. The karyotype of *Th. intermedium* revealed two sets of seven chromosomes that were longer than the third set of seven chromosomes, which was the **St** genome. When *Th. intermedium* × *Th. bessarabium* hybrid was analyzed, seven long chromosomes were attributed to *Th. bessarabium* (**J^b^** genome); 14 intermediate (**J^e^**) and seven short chromosomes (**St**) were from *Th. Intermedium*. It was shown in this study that when **J^b^** was present with **St**, the ratio between the longest and shortest chromosome was about 2.3. The ratio was around 1.8 when **J^e^** and **St** were present in combination. This ratio was 1,3 within the **J^b^** genome.

In 1992, Liu and Wang [36] had given the genome symbol **J^b^J^e^** to both *Th. junceiforme* and *Th. sartorri*, but noted variations in the satellite number and size as well as C-banding.

A few years earlier, Pienaar et al. [37] studied the genome relationships in *Thinopyrum* species. Genome constitution was determined in *Th. scirpeum* as **J^e^J^e^**, *Th. distichum* **J^d^J^d^**, *Th. junceiforme* **J_1_J_2_**.

Using in situ hybridization and molecular markers, Wang and Zhang [14] first used the **St** genomic DNA in a GISH study of two translocation lines involving *Th. intermedium* that conferred the resistance to either wheat streak mosaic virus or barley yellow dwarf virus. The presence of **St** chromatin in these two translocation lines CI17766 and TC14 was substantiated also by **St**- and **E**-specific RAPD cloned marker OPB08_525_ and OPC03_340_, respectively.

In Canada, Chen et al. (1998) [15] analyzed both *Th. intermedium* and *Th. ponticum* using the genomic DNA from *Th. elongatum* (Host) D.R. Dewey (genome **E**, 2*n* = 14), *Th. bessarabicum* (Savul. & Rayss) Á. Löve (genome **J**, 2*n* = 14), and *Ps. strigosa* (M. Bieb.) Á. Löve (genome **S**t, 2*n* = 14). They gave *Th. intermedium* the genome designation **JJ^s^S**, where **J** was related to the **E** genome of *Th. elongatum* and the **J** genome of *Th. bessarabicum*, the **S** genome was homologous to the **S** genome of *Ps. strigosa*, while the **J^s^** genome referred to modified **J**- or **E**-type chromosomes distinguished by the presence of **S** genome-specific sequences close to the centromere. However, the **J^s^** genome was not composed of fourteen, but only nine to eleven chromosomes.

Later, Kishii et al. (2005) [16] showed that *Th. intermedium* contains three kinds of genomes: **St**, **E/J**, and the third genome might be close to the **V** genome. However, PCR analysis disconfirmed the presence of the present-day **V** genome in *Th. intermedium*, but showed some similarity in the **R** genome. Thus, the genomic formula of *Th. intermedium* was tentatively re-designated as **StJ^s^**(**V-J-R**)**^s^**. Then in 2011, the study of Mahelka et al. [17] further complicated the genome constitution of *Th. intermedium* by GISH and nuclear GBSSI sequences suggesting that present-day genera *Pseudoroegneria, Dasypyrum, Taeniatherum, Aegilops,* and *Thinopyrum* were involved in the evolution of IWG. The DNA of the former two genera consistently hybridized to two genomes of *Th. intermedium*, but the third genome could be hybridized by those of the other three genera.

### 3.2. Current Studies on Thinopyrum intermedium and Related Species

The ten accessions of IWG studied here had the same genome constitution **J^r^J^r^J^vs^J^vs^StSt**, indicating that IWG populations collected from a wide range of regions (Table 1) had the same evolutionary end product even though it might have gone through different pathways, i.e., different diploid *Pseudoroegneria* species such as *P. libanotica*, *P. stipieforlia*, or *P. strigosa* could be hybridized by different tetraploid *Thinopyrum* species such as *Th. junceiforme* or *Th. sartorii*. The multiplex oligonucleotide FISH patterns of these IWG accessions, however, revealed variability among and within accessions. These variations could be attributed to the outcrossing nature of this species [38] and multiple hybridization from different sources [17].

Understanding the relationships among **J, St,** and **ABD** of wheat, Wang’s laboratory [14,39,40] first used the genomic DNA of **St** as a probe for the GISH study of wheat hybrids with *Th. ponticum* and *Th. intermedium*. The use of **St** genomic DNA as a GISH probe for the detection of **J** chromosomes or chromosome fragments in hybrid derivatives of wheat × *Thinopyrum intermedium* was endorsed as a “landmark approach’ for tracing the introgression of **J** chromatin into wheat [41]. However, our findings of St_2_-80 hybridization signals at distal ends of chromosomes in *Thinopyrum* species (Figure 1, Figure 2, Figure 3, Figure 4, Figure 5 and Figure 6) might set the limitation of this approach of GISH detection for useful alien genes. If the genes of interest were located in the interstitial region of **J** chromosomes, **St** genomic probe would likely fail to detect the introgression of alien chromatin.

Because IWG had been proposed to have a haplome formula of **J^e^J^e^St** [13], naturally occurring tetraploid *Thinopyrum* species having the genome composition **J^b^J^b^**, **J^b^J^e^** or **J^e^J^e^** would be a potential progenitor of IWG. Indeed, tetraploid *Th. junceiforme* and *Th. sartorii* previously shown to have the haplome composition **J^b^J^e^** [36] were found to have the **J^vs^J^r^** portion of **J^vs^J^r^St** in IWG (Figure 3 and Figure 4). Thus, these two tetraploid *Thinopyrum* species are now the candidates for a progenitor of IWG. One the other hand, *Th. scirpeum* that is **J^r^J^r^** (Figure 5) could be excluded from the candidate progenitors of *Th. intermedium*.

One easy way to ascertain the presence of the **St** genome along with the **J** genome is to determine the chromosome length ratio between the longest and shortest chromosomes in the studied species. The ratio will be greater than 2.3 when **J^b^** and **St** are present; about 1.8 when **J^e^** and **St** are present; and about 1.3 when **St** is absent [35]. The ratio around 2.3 was observed in karyotypes of *Th. intermedium* (Figure 1 and Figure 2) where **St** was present. The ratio around 1.3 was found in *Th. junceiforme, Th. sartorii,* and *Th. scirpeum* (Figure 3, Figure 4 and Figure 5), indicating the absence of **St.** The autotetraploidy of *Th. scirpeum* [37] is now confirmed by its **J^r^J^r^J^r^J^r^** genome composition.

Reporting the development of pDb12H, Yang et al. stated that this FISH probe detected all chromosomes of *Da. breviaristatum* (see Figure 6 of [21]), but the FISH signal was not detectable in *Da. villosum* chromosomes in diploid accessions [21]. However, they did not provide the photographic evidence for the presence or absence of pDb12H in *Da. villosum*. Our Appendix A clearly shows that pDb12H could only weakly hybridize the pericentromeric regions of eight chromosomes of *Da. villosum*. This result supports the conclusion that *Da. villoum* and *Da. breviaristatum* are distinct, deserving the designation of genome symbols **V^v^** and **V^b^**, respectively [42,43]. It also substantiates the conclusion that the present-day **V** genome of *Da. villosum* is not the **J^vs^** genome [23].

Most importantly, the observations that genomic DNA of *Th. elongatum* or *Th. bessarabicum* hybridized only to the same 14 chromosomes in IWG (Figure 7A,B) indicate that IWG had only one **J** genome. This observation is unique and different because the same root-tip cell was used in sequential GISH experiments. All other studies mentioned earlier had different root-tip cells used for GISH using different probes. Our finding in this study supports that reported for *Th. junceiforme* [44]. Furthermore, the **J^vs^** genome could be hybridized by either genomic DNA of *Da. villosum* or oligo probe pDb12H (Figure 8), indicating that **J^vs^** was likely a progenitor of *Da. breviaristatum* and *Da. villosum*. Based on Appendix A, it can be inferred that the **V^v^** genome of the latter had a much lower copy number of the pDb12H sequence than **V^b^** of *Da. breviaristatum*. When more evidence becomes available, the genome symbol **J^vs^** might be changed to **V^b^**.

### 3.3. Future Studies on Thinopyrum intermedium and Related Species Needed

The recent development of the precise identification of *Th. intermedium* chromosome compliment [45] and chromosome-specific bulked oligonucleotides for identifying **E**-genome chromosomes in both *Th. bessarabicum* and *Th. elongatum* [46] would be useful in future studies on all *Thinopyrum* species and wheat × *Thinopyrum* hybrid derivatives.

In addition, a polyhaploid of *Th. intermedium* (2n = 3x = 21, **J^vs^J^r^St**) and the hybrid between *Th. intermedium* and different *Pseudoroegneria* species (2n = 4x = 28, **J^vs^J^r^StSt**) should be made. Then, oFISH using pDb12H [21,22], St_2_-80 [20], and pMD232-500 [18] should be carried out on both root-tip cells and pollen mother cells of these plants. The results from oFISH of pollen mother cells would reveal whether the two genomes **J^vs^** and **J^r^** are capable of high pairing. If the two genomes can pair to form four or more bivalents, their genome symbols would stay. If the two genomes cannot form any bivalent, the **J^vs^** might have to be changed to **V**.

## 4. Materials and Methods

### 4.1. Plant Materials

*Thinopyrum intermedium* (2n = 6x = 42), *Th. junceiforme* (2n = 4x = 28), *Th. sartorii* (2n = 4x = 28), *Th. scirpeum* (2n = 4x = 28), *Th. elongatum* (2n = 2x = 14, **E^e^E^e^**), *Th. bessarabicum* (2n = 2x = 14, **JJ** or **E^b^E^b^**), *Pseudowroegneria spicata* (2n = 2x = 14, **StSt**) having the PI numbers were kindly provided by the Germplasm Resource Information Network (GRIN) of United States Department of Agriculture (Table 1). *Dasypyrum villosum* (2n = 2x = 14, **VV**) were obtained from Prof. Xingfeng Li, College of Agronomy, Shandong Agricultural University. All plant materials were maintained through selfing at the Tai’an Subcenter of the National Wheat Improvement Center, Tai’an, China.

### 4.2. DNA Extraction and Probe Preparation

The CTAB method was used to extract total genomic DNA from *Th. elongatum*, *Th. bessarabicum*, *Ps. strigose*, *Da. villosum*. Two oligonucleotide probes used for the FISH studies were St_2_-80 [20] and pDb12H [21,22]. pDbH12 could serve as a cytogenetic marker for tracing chromatin from the **V^b^** genome in wheat–alien introgression lines. St_2_-80 is a potential and useful FISH marker that can be used to distinguish **St** and other genomes in Triticeae. Fluorescent signals of *Th. elongatum* (**E^e^** = **J^e^**), *Th. bessarabicum* (**E^b^** = **J^b^**), and *Ps. strigose* (**St**) genomic DNA as well as St_2_-80 were labeled with Texas-red-5-dCTP, while pDb12H and *Da. villosum* (**V**) genomic DNA were labeled with fluorescein-12-dUTP using the nick translation method. Oligonucleotides (synthesized by Sangon Biotech, Shanghai, China) pSc119.2-1 and (GAA)_10_ were labeled with 5′-FAM (5-carboxyfluorescein), while AFA-3, AFA-4, pAs1-1, Pas1-3, pAs1-4, and pAs1-6 were labeled with 5′-TAMRA (5-carboxytetramethylrhodamine) as described by Du et al. 2017 [47].

### 4.3. Chromosome Preparation and GISH, FISH Protocol

Fresh root-tip cells (RTCs) collected from germinating seeds were treated with 1.0 MPa nitrous oxide (N_2_O) for 2 h [48] and then immersed in 90% glacial acetic acid. RTC slides were prepared according to the procedure used in Prof. Han’s laboratory [49]. Once spread chromosomes were found under the phase-contrast microscope, the slides were put in liquid nitrogen for freezing and the coverslips were removed. Then slides were dehydrated in ethanol and dried at room temperature. After the above steps, the slides were subjected to FISH. The procedures of GISH, FISH, and signal detection were conducted according to the method of Du et al. (2017) [47] and He et al. (2017) [50]. The probe labelling, denaturation, image capture, and data processing were described in Cui et al. (2019) [51]. Images were collected using a NIKON eclipse Ni-U fluorescence microscope; the images were processed using NIS-Elements BR 4.00.12 software.

## 5. Conclusions

In this study, we found that using two oligonucleotides pDb12H and St_2_-80 as probes in FISH was sufficient to distinguish the three genomes **J^vs^**, **J^r^**, and **St** in intermediate wheatgrass. In addition, variations in the outcrossing *Th. intermedium* were visible in multiplex oFISH both among and within accessions. *Th. junceiforme* and *Th. sartorii* were probable progenitors of IWG. **J^r^** is related to either **J^e^** of *Th. elongatum* or **J^b^** of *Th. bessarabicum*. Finally, the **J^vs^** genome could be the progenitor of present-day **V^b^** and **V^v^**; thus, this genome symbol might be changed to **V** when enough evidence becomes available.

## Figures and Tables

**Figure 1 plants-12-03705-f001:**
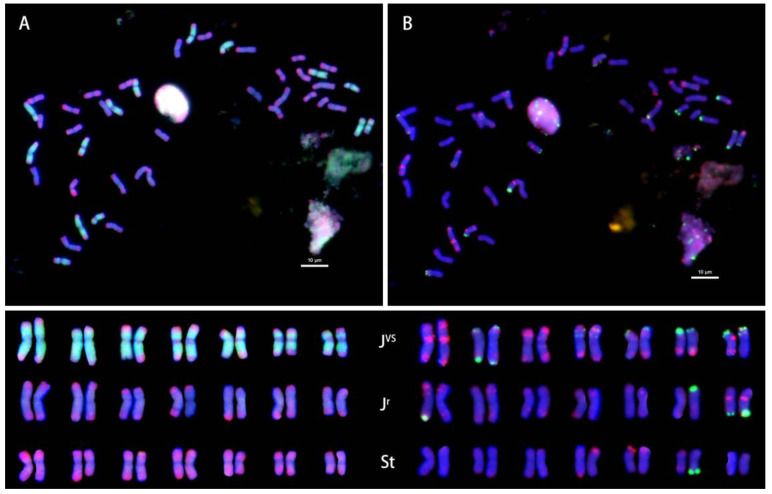
FISH characterized chromosomes (**top**: (**A**,**B**)) and karyotypes (**bottom**) of *Thinopyrum intermedium* PI 325190. (**Bottom**) left side: probed with pDb12H (green) and St_2_-80 (red); right side: probed with multiplex oligonucleotides, including pSc119.2-1, (GAA)_10_ (green), AFA-3, AFA-4, pAs1-1, Pas1-3, pAs1-4, and pAs1-6 (red), Scale bar: 10 μm.

**Figure 2 plants-12-03705-f002:**
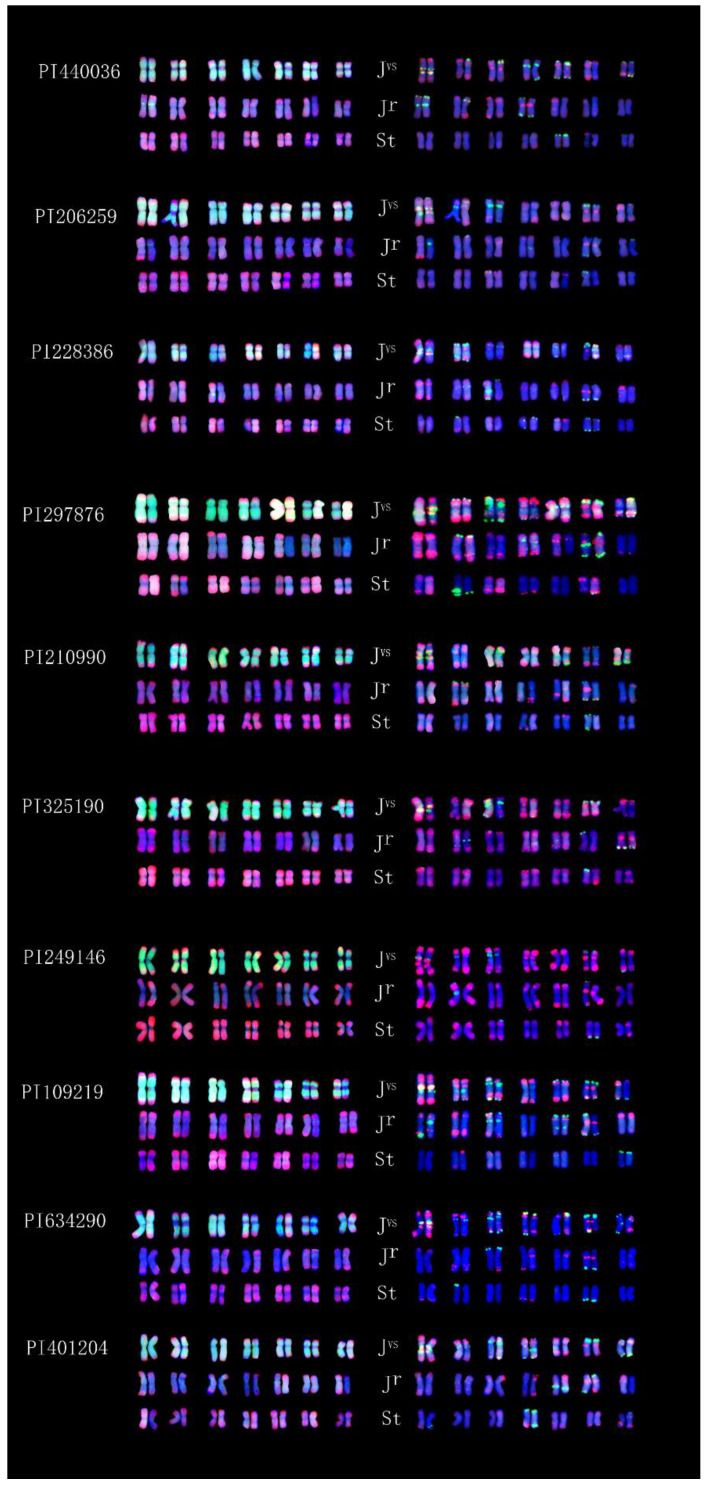
Karyotypes of ten accessions of *Thinopyrum intermedium* (**top** to **bottom**) PI 206259, PI 210990, PI 228386, PI297876, PI 325190, PI 440096, PI 249146, PI 109219, PI 634290, and PI 401204. Left side: probed with pDb12H (green) and St_2_-80 (red). Right side: probed with multiplex oligonucleotides, including pSc119.2-1, (GAA)_10_ (green), AFA-3, AFA-4, pAs1-1, Pas1-3, pAs1-4, and pAs1-6 (red).

**Figure 3 plants-12-03705-f003:**
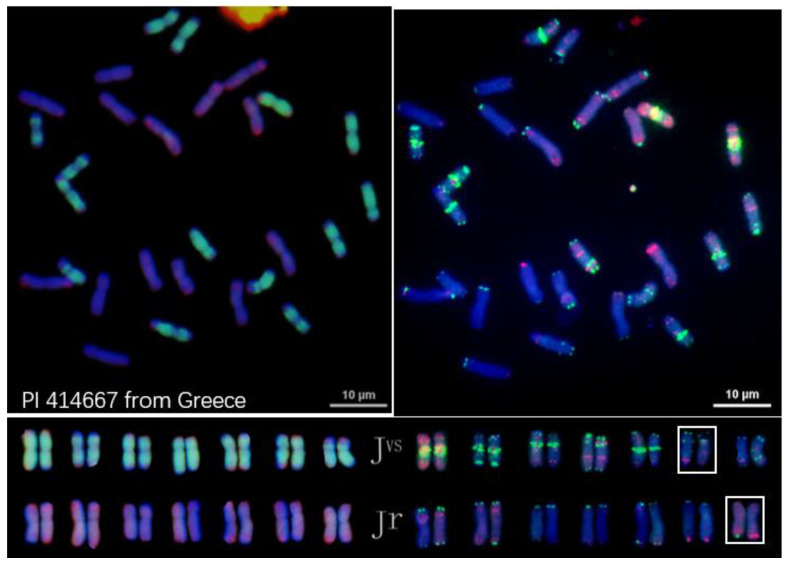
FISH characterized chromosomes and karyotypes of *Thinopyrum junceiforme* PI 414667. Left side: probed with pDb12H (green) and St_2_-80 (red). Right side: probed with multiplex oligonucleotides, including pSc119.2-1, (GAA)_10_, AFA-3, AFA-4, pAs1-1, Pas1-3, pAs1-4, and pAs1-6. Variations in FISH signals between homologous chromosomes are detected (in rectangular boxes).

**Figure 4 plants-12-03705-f004:**
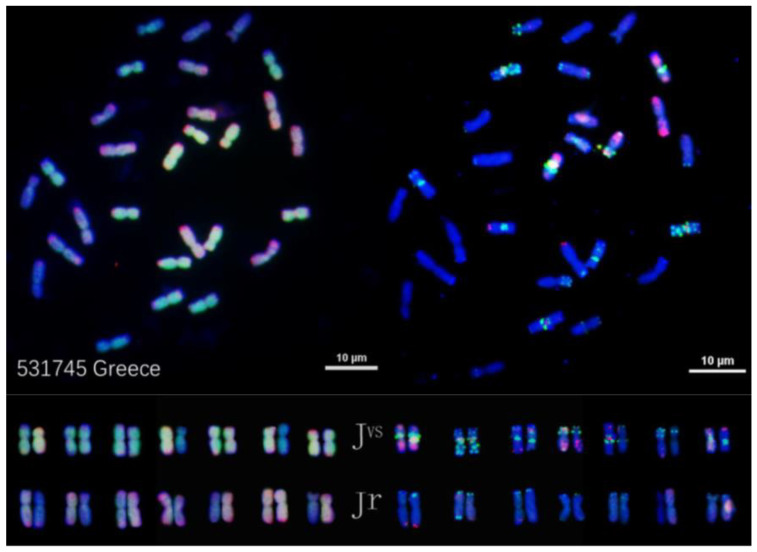
FISH characterized chromosomes (**top**) and karyotypes (**bottom**) of *Thinopyrum sartorii* PI 531745. Left side: probed with pDb12H (green) and St_2_-80 (red). Right side: probed with multiplex oligonucleotides, including pSc119.2-1, (GAA)_10_, AFA-3, AFA-4, pAs1-1, Pas1-3, pAs1-4, and pAs1-6.

**Figure 5 plants-12-03705-f005:**
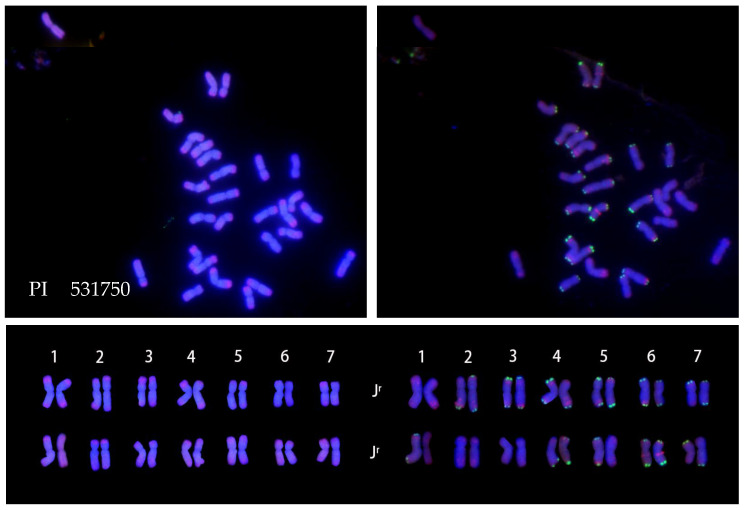
FISH characterized chromosomes (**top**) and karyotypes (**bottom**) of *Thinopyrum scirpeum* PI 531750. Left side: probed with pDb12H (green) and St_2_-80 (red). Right side: probed with multiplex oligonucleotides, including pSc119.2-1, (GAA)_10_, AFA-3, AFA-4, pAs1-1, Pas1-3, pAs1-4, and pAs1-6.

**Figure 6 plants-12-03705-f006:**
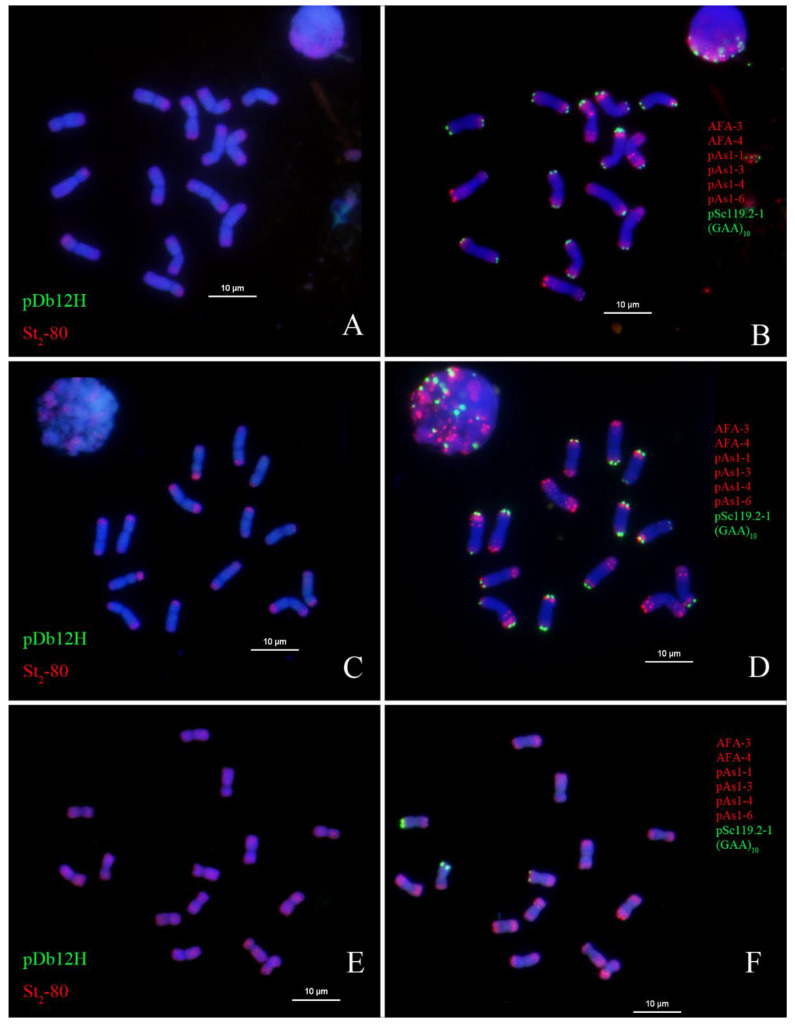
Oligonucleotide FISH of *Thinopyrum elongatum* (**A**), *Th. bessarabicum* (**C**), and *Pseudoroegneria strigose* (**E**), probed with pDb12H (green) and St_2_-80 (red). Oligonucleotide FISH of *Thinopyrum elongatum* (**B**), *Th. bessarabicum* (**D**), and *Pseudoroegneria strigose* (**F**), probed with multiplex oligonucleotides, including pSc119.2-1, (GAA)_10_ (green), AFA-3, AFA-4, pAs1-1, Pas1-3, pAs1-4, and pAs1-6 (red).

**Figure 7 plants-12-03705-f007:**
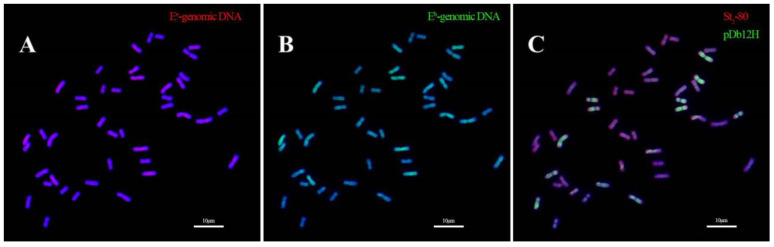
*Thinopyrum intermedium* probed with genomic DNA of *Th. elongatum* (**A**), *Th. bessarabicum* (**B**), and oligo pDb12H and St_2_-80 (**C**). All three experiments were conducted with the same root-tip cell, Scale bar: 10 μm.

**Figure 8 plants-12-03705-f008:**
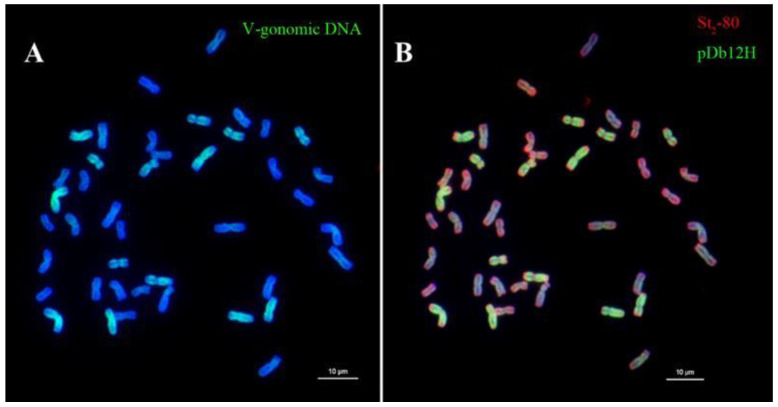
Chromosomes of *Thinopyrum intermedium* probed with (**A**) genomic DNA of *Dasypyrum villosum* and (**B**) oligos pDb12H (green) and St_2_-80 (red).

**Table 1 plants-12-03705-t001:** Plant materials used in this study.

Species	ID	Chr Number	Origin	Note
*Thinopyrum intermedium* (Host) Barkworth & D. R. Dewey	PI 109219	42	District of Columbia, United States	
	PI 206259	42	Turkey	
	PI 210990	42	Afghanistan	
	PI 228386	42	Iran	
	PI 249146	42	Portugal	
	PI 297876	42	Former, Soviet Union	
	PI 325190	42	Stavropol, Russian Federation	
	PI 401204	42	Iran	
	PI 440036	42	Kazakhstan	
	PI 634290	42	Krym, Ukraine	
*Th. junceiforme* (A. & D. Löve) A. Löve	PI 414667	28	Greece	listed as *Thinopyrum junceum* (L.) Á. Löve
*Th. sartorii* (Bioss. & Heldr.) Á. Löve	PI 531745	28	Greece	
*Th. scirpeum* (C. Presl) D. R. Dewey	PI 531750	28	Greece	
*Th. bessarabicum* (Savul. & Rayss) A. Löve	PI 531712	14	Estonia	
*Th. elongatum* (Host) D. R. Dewey	PI 340063	14	Turkey	
*Pseudoroegneria spicata* Pursh) Á. Löve	PI 563869	14	Oregon, USA	
*Dasypyrum villosum* (L.) Candargy		14		From X-F Li’s collection

## Data Availability

All figures in this article are available for use without restrictions.

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
