# Peer review of "Genome Analysis of Thinopyrum intermedium and Its Potential Progenitor Species Using Oligo-FISH"

_plants, 2023, doi:10.3390/plants12213705_

Round 1

Reviewer 1 Report

The authors can find the comments in the attached file.

The comments about the quality of English can be found in the attached file.

Author Response

Thank you for your thorough review of our submitted manuscript and valuable suggestions that contributed to true improvement of the original manuscript. 

Reviewer 2 Report

In this manuscript, “In situ hybridization studies of Thinopyrum intermedium and its progenitor species”, were used as probes in fluorescence in situ hybridization (FISH) and multiplex oligonucleotides FISH were used to investigate the karyotype composition of Th. intermedium and its relative diploid s pecies. The results are interesting and provide new insights into the genome constitution of Th. intermedium and its evolution.

In my opinion, this manuscript was suitable for publication in “BMC Plant Biology” after minor revision. Some suggestions are as follows:

1). In Fig.2, ‘PI’ should be added before the PI number.

2). Chromosome in Fig. 5 was not as clear as others, it should it be replaced.

3). There were no with multiplex oligonucleotides FISH results of Thinopyrum elongatum and Th. bessarabicum, it should be added.

Author Response

Thank you for your review of our original manuscript and valuable suggestions for improvements in the revised manuscript. 

Reviewer 3 Report

In this MS, authors explained oligoFISH of Th. intermedium and some of its progenitor species. I see the results and discussion parts are written properly. However, I have some suggestions to improve this MS to make it publishable as:

1. Abstract is not reflecting the summery of your total results especially Fig. 6, 7, and 8. Additional sentences are needed. you may think of removing lines 13-14.

2. line 31-32 are reflection of line 13-14. revise those

3. In the introduction, no explanation of oFISH and GISH and their ways of function. please mention this in a paragraph.

4. All the figures are not self explanatory. need some clarification like labelled with .... (green),---(red)

5. No explanation of image capture at what magnification and software used to prepare data.

6. In the methods and materials, brief discussion on probe labelling, denaturation, image capture are needed.

English are simple and fine

Author Response

Thank you for your review of our original manuscript. Your suggestions contributed to the greatly improved manuscript that should be acceptable for publication in the special issue of Plants.

Round 2

Reviewer 1 Report

I find the revised version of the manuscript acceptable for publication.

I just noticed two fragments that need editing:

-        lines 202-216 have to be justified

-        lines 371-372 the font is different

Reviewer 3 Report

I am satisfied with the revised manuscript